# Differences in the Estimation of Wildfire-Associated Air Pollution by Satellite Mapping of Smoke Plumes and Ground-Level Monitoring

**DOI:** 10.3390/ijerph17218164

**Published:** 2020-11-05

**Authors:** Raj P. Fadadu, John R. Balmes, Stephanie M. Holm

**Affiliations:** 1Joint Medical Program, University of California Berkeley School of Public Health and University of California San Francisco School of Medicine, Berkeley, CA 94704, USA; john.balmes@ucsf.edu; 2Division of Environmental Health Sciences, School of Public Health, University of California Berkeley, Berkeley, CA 94704, USA; 3Department of Medicine, University of California San Francisco, San Francisco, CA 94143, USA; stephaniemholm@berkeley.edu; 4Division of Epidemiology, School of Public Health, University of California Berkeley, Berkeley, CA 94704, USA; 5Western States Pediatric Environmental Health Specialty Unit, San Francisco, CA 94143, USA

**Keywords:** air pollution, wildfires, exposure assessment, environmental epidemiology, environmental health

## Abstract

Wildfires, which are becoming more frequent and intense in many countries, pose serious threats to human health. To determine health impacts and provide public health messaging, satellite-based smoke plume data are sometimes used as a proxy for directly measured particulate matter levels. We collected data on particulate matter <2.5 μm in diameter (PM_2.5_) concentration from 16 ground-level monitoring stations in the San Francisco Bay Area and smoke plume density from satellite imagery for the 2017–2018 California wildfire seasons. We tested for trends and calculated bootstrapped differences in the median PM_2.5_ concentrations by plume density category on a 0–3 scale. The median PM_2.5_ concentrations for categories 0, 1, 2, and 3 were 16, 22, 25, and 63 μg/m^3^, respectively, and there was much variability in PM_2.5_ concentrations within each category. A case study of the Camp Fire illustrates that in San Francisco, PM_2.5_ concentrations reached their maximum many days after the peak for plume density scores. We found that air pollution characterization by satellite imagery did not precisely align with ground-level PM_2.5_ concentrations. Public health practitioners should recognize the need to combine multiple sources of data regarding smoke patterns when developing public guidance to limit the health effects of wildfire smoke.

## 1. Introduction

The number and intensity of wildfires in the United States (U.S.) and elsewhere around the world have been increasing over the past several years and are projected to further increase with progressive climate change [1,2,3,4]. The recent wildfires in the Western U.S. have burned over 6 million acres of land, and the resulting smoke has spread far distances to the Eastern U.S. and even Northern Europe [5]. In addition, wildfires in Brazil and Australia recently garnered international media attention for their devastating effects on global communities, exposing millions of people to hazardous levels of air pollutants [6,7]. Wildfires are a significant source of particulate matter <2.5 μm in diameter (PM_2.5_) [8,9], which has been found to be associated with negative respiratory, cardiovascular, neurologic, skin, and metabolic health effects [10,11,12,13]. Climate change is predicted to increase the number of high-pollution smoke days in the Western U.S., which may cause excess future hospitalizations for respiratory disease [14]. As climate change progresses, the impacts of smoke from wildfires on public health are becoming increasingly larger.

Prior research suggests that air pollution exposure attributable to wildfires has effects on both individuals and healthcare systems. These effects are likely due to its associations with both a wide range of harmful effects on human health and increased hospitalization rates, which have implications for public health practice and policy [15,16,17,18,19]. Many epidemiologic studies demonstrate increased risk for adverse health outcomes (cardiovascular, respiratory, and mortality) in association with wildfire smoke exposure [20,21,22,23,24,25].

These studies use varying approaches to exposure assessment, including concentrations of air pollutants measured by ground monitors [20,21,22,23], air quality indices created by government agencies [20], and smoke plume density scores determined by satellite imagery [24]. In addition, some researchers have combined spatiotemporal data sets to perform sophisticated modeling of PM_2.5_ exposure from wildfire smoke using data-adaptive machine learning and coupled empirical and deterministic models [26,27,28]. Each approach has its strengths and weaknesses, which affect the interpretation of study findings and the translation of research to public health practice.

Public messaging about changes in air quality due to wildfire smoke in the U.S. currently depends on ground-level concentrations of PM_2.5_ to calculate an air quality index, but satellite-based monitoring of smoke plumes to describe the haziness of atmospheric conditions is also available. These exposure datasets are more likely to be used by public health practitioners than complex metrics, such as aerosol optical depth and fire radiative power observations, because data on pollutant concentrations from ground-level monitors and smoke density scores from satellite imagery are open-access, easily accessible, and straightforward to interpret. Both have been integrated into the U.S. Environmental Protection Agency’s newly updated fire and smoke map (fire.airnow.gov), which is used to provide information on air quality during wildfire events to the national public [29]. Some investigators have attempted to combine data on PM_2.5_ concentrations and smoke density to improve wildfire smoke exposure assessment [23,24]. Here, we present our experience comparing ground-level monitoring and satellite imagery data to demonstrate previously unreported disparities in exposure assessment methods, highlighting nuances of characterizing wildfire-associated air pollution that can affect public communication and research about wildfire smoke exposure.

## 2. Materials and Methods

We assessed the impacts of wildfires on air quality in the San Francisco Bay Area during the 2017 and 2018 wildfire seasons, from 1 June to 30 November of both years (N = 366 days, ranging from 346–366 observations per monitor due to some days when measurements were not available for select monitors). To do so, we collected air pollution data from two publicly available sources: PM_2.5_ concentration data from Bay Area Air Quality Management District (BAAQMD, San Francisco, CA, USA, www.baaqmd.gov) monitoring stations and smoke plume density data from the National Oceanic and Atmospheric Administration Hazard Mapping System (NOAA HMS, Suitland, MD, USA) for Fire and Smoke product (www.ospo.noaa.gov/Products/land/hms.html).

We gathered data on daily 24-h average and daily maximum hourly average PM_2.5_ concentrations (μg/m^3^) from the BAAQMD. We included all the BAAQMD ground-level monitors in the Bay Area that measured PM_2.5_ concentrations during 2017 and 2018, which yielded a total of 16 stations in both urban and non-urban settings. The locations of these stations are displayed in Figure 1. The BAAQMD validates their data through audits for quality assurance that meet the U.S. Environmental Protection Agency’s requirements.

In addition, we gathered data on daily maximum smoke plume density score, ranging from 0 to 3, for the zip code where each BAAQMD air-quality monitoring station is located from the Hazard Mapping System (HMS), which integrates data on the geographic spread of smoke plumes from satellite images. Fire detection zones and smoke analyses are updated several times per day. The estimates are separated into four categories—no wildfire plume (0); and light (1), medium (2), and heavy smoke (3)—that describe air quality for a given geographic area. A score of 0 refers to no wildfire smoke contributing to air pollution, and scores 1–3 theoretically reference PM_2.5_ concentrations attributable to wildfire smoke: light (1), 0–10 μg/m^3^; medium (2), 10.5–21.5 μg/m^3^; and heavy (3), ≥22 μg/m^3^ [30,31].

Statistical analyses and visualizations were performed with the R statistical programming language, version 3.6.3, using the following packages: boot, DescTools, dplyr, ggplot2, lubridate, perm, SF, and tmap. Data on daily PM_2.5_ concentrations and smoke plume density scores at all 16 locations were collectively analyzed. Analyses using both the daily 24-h average and daily maximum hourly average PM_2.5_ concentrations yielded similar results, so we only report results using the latter. The PM_2.5_ concentrations occurring during the four smoke plume categories were compared using the Jonckheere–Terpstra test, a ranks-based test that non-parametrically assesses differences among ordered groups. One-sided alternative hypotheses were used to specifically test whether increasing smoke plume score corresponded with increasing PM_2.5_. Post-hoc testing was performed using both permutation tests and bootstrapped estimates. Bootstraps were calculated with 95% confidence intervals for the differences in the medians calculated using the percentile technique.

## 3. Results

### 3.1. Analysis of the Full Wildfire Seasons

When analyzing data from the entire 2017–2018 wildfire seasons in California, we found that there is much variability in maximum daily PM_2.5_ concentrations within each smoke plume category (Figure 2). Categories 0, 1, and 2 have many outliers at the upper end of their distribution of PM_2.5_ concentrations, which indicates high PM_2.5_ concentrations on days assigned a smoke plume score less than 3. Outliers were distributed across monitoring sites, with no sites particularly prone to outlier values. The median values for daily maximum hourly average PM_2.5_ concentration for smoke plume density categories 0, 1, 2, and 3 are 16, 22, 25, and 63 μg/m^3^, respectively. These findings do not align well with the theoretical target PM_2.5_ concentration range referenced for smoke plume categories 1–3 (0–10; 10.5–21.5; ≥22) [30,31]. Despite this variability, a test for trend indicates a significant increasing trend in PM_2.5_ when considering all four categories (*p* < 0.001) and only the categories where a smoke plume is present (categories 1, 2, and 3; *p* < 0.001).

The estimated median differences in the daily maximum PM_2.5_ concentrations between days with smoke plumes (ratings of 1, 2, or 3) and days without smoke plumes (rating of 0) are all positive. Similarly, the PM_2.5_ concentrations on days with a smoke plume rating of 3 are generally higher than those on days with a lower rating, though there is great variance. For instance, the estimated difference in the median PM_2.5_ concentrations on days with a smoke plume rating of 3 compared to days with a rating of 2 is 35.1 μg/m^3^ (95% CI: 13.5, 51.0). The estimated difference in median PM_2.5_ concentrations between days with smoke plume rating of 2 and 1 is much smaller: 3.5 μg/m^3^ (95% CI: 1.0, 6.0).

### 3.2. Case Study of the Camp Fire Smoke Reaching San Francisco

Focusing on the 2018 Camp Fire as an example, the time-series plot (Figure 3) demonstrates that PM_2.5_ concentrations and smoke plume density scores in the city of San Francisco started to significantly increase around 8 November 2018 and decreased by 21 November 2018: a 2-week period. Both of these increases are associated with high levels of air pollution that can be attributable to the wildfire. However, there is a significant disparity in PM_2.5_ characterization by ground-level monitoring and the HMS during this time frame. While the daily maximum PM_2.5_ concentration spikes to its maximum during the second week, the smoke plume density score is highest at the beginning of this time frame and remains smaller for the rest of the period. Notably, the PM_2.5_ concentrations at this location are substantially higher than the intended smoke plume 3 values for the entire interval, and air pollution assessment via ground-level monitoring of PM_2.5_ concentration and satellite imagery of smoke plumes differ temporally. Use of the smoke plume data alone could give the false impression that wildfire smoke exposure posed a higher health risk early in the event.

## 4. Discussion

Different exposure assessment methodologies have been used to communicate wildfire smoke hazards to the public, but these convey different information about smoke-related air pollution. Ground-leveling monitoring and satellite imagery are two commonly used pollution assessment techniques. Both are reported with high spatial variability on the AirNow fire and smoke map, which communicates information about fire and air quality to the U.S. public [29]. Some investigators have examined the impacts of wildfire-associated fine particulate matter by using the NOAA HMS smoke plume density scores to categorically assess exposure [23]. However, in our analysis of PM_2.5_ arising from the full 2017–2018 California wildfire seasons, we demonstrate that the ground-level PM_2.5_ concentrations recorded by BAAQMD air monitors in the San Francisco Bay Area did not closely correlate with the NOAA HMS smoke plume density scores for those same locations. While there was a significant increasing trend in PM_2.5_ concentrations across smoke plume categories, there was substantial variability within each of the categories as well. The extent of overlap in the range of PM_2.5_ concentrations for each smoke plume category is a good reminder that using smoke plume data alone may not be a good proxy for quantifying atmospheric PM_2.5_ concentrations. This issue is further illustrated by our analysis of the 2-week period of the Camp Fire, during which the smoke plume density score was highest at the beginning of the first week and then decreased, while the PM_2.5_ concentration reached its maximum in the second week.

These differences highlight the fact that the two sources of data understandably characterize wildfire-associated air pollution differently. The NOAA HMS uses visible band satellite imagery with a 4-km resolution to ascribe a density score for smoke plumes [32]. These scores may not precisely or accurately describe the concentration of PM_2.5_ for the corresponding geographic area because they measure anything that is blocking light, including particles of many sizes, not only PM_2.5_. Misclassification of exposure between the ground level and what is observed higher in the atmosphere can occur because the vertical position of the smoke plume is often unknown on satellite imagery [15]. Additionally, many factors can influence the spread of smoke plumes and particles of different sizes. The presence of outliers in each smoke plume category could be explained by wind, temperature, relative humidity, and coastal marine fog. These climatic and geographic factors can affect the spread of wildfire smoke from the source to ground-level sensors, leading to differences in estimated pollution levels. For the example of the Camp Fire, we hypothesize that while the smoke was heaviest at the beginning of the fire, PM_2.5_ may have accumulated in San Francisco later. Thus, ground-level measurements are an important supplement to smoke plume data, as decreasing smoke plume scores alone are insufficient to suggest that health risks are decreasing. Similarly, the absence of a visible smoke plume may not be sufficient to rule out wildfire smoke as a contributor to elevated particulate matter levels. Guidance provided to the public, including those with chronic heart or lung conditions that may be exacerbated by exposure to wildfire smoke, should incorporate both sources of exposure information.

This study’s findings have implications for future public health research related to climate change and wildfires. As researchers continue to study the effects of wildfire-associated air pollution on morbidity and mortality, they must understand that their results should be interpreted within the limitations of their chosen exposure assessment techniques. For example, many environmental epidemiology studies use exposure lag analysis to account for cumulative or delayed impacts of wildfire smoke exposure on health outcomes [17,20,23,33,34]. The associations that researchers find between exposure and outcomes at various lag times can be affected by the use of specific exposure assessment methods; our analysis of the Camp Fire shows that the time course of the exposure patterns can vary by exposure method over the duration of the same wildfire event. Further research on exposure assessment should explore differences in how other techniques, such as aerosol optical depth and fire radiative power observations, estimate wildfire-associated air pollution.

This study has some limitations. We analyzed wildfire-associated air pollution using data from ground-level monitoring stations located in one geographic area in Northern California, which may limit the generalizability of our results. Our findings demonstrate that ground-level measurements of PM_2.5_ did not always characterize air quality in a similar manner to data from satellite imagery, and additional studies are needed to see whether these results hold true in other cases of wildfires in different geographic areas. Additionally, it is possible that the measurements of PM_2.5_ concentrations could have been impacted by factors other than wildfires, though the high increases above background levels that we observed consistently correspond to the timing of wildfires.

Quantifying wildfire-associated particulate matter pollution is challenging, and two commonly used sources of particulate matter data in public health research are satellite imagery and ground monitor measurements. These data are publicly available and commonly used by public health practitioners to inform policy and decisions regarding health risks and public safety. Our findings suggest that NOAA smoke plume density scores did not accurately and consistently reflect ground-level PM_2.5_ concentrations in the Bay Area during the 2017–2018 California wildfire seasons. Therefore, ground monitors can effectively capture surface level concentrations of PM_2.5_, while the HMS may be better suited for characterizing the spatial distribution of smoke plumes from wildfires. This distinction is critical because low smoke plume density scores on a given day may suggest that the air quality is healthy enough for prolonged outdoor activity when particulate pollution is actually still high. Accurate risk communication about hazardous air quality conditions during wildfires is important because exposure to PM_2.5_ is associated with many harmful negative health effects [35,36,37]. Public health researchers and practitioners should be aware of the limitations of different wildfire-associated particulate matter estimation methodologies when interpreting data for important policy decisions and communicating information on air quality to the public.

Due to these limitations, we support the refinement of combined air pollution exposure assessment methods, including the integration of measured and modeled atmospheric data. As examples, the Community Multiscale Air Quality (CMAQ) model is a multipollutant, deterministic modeling system with temporal and spatial flexibilities, and the Forum for Air Quality Modelling in Europe (FAIRMODE) is a network that strives to harmonize the use of air pollution assessment practices [38,39]. More robust data integration methods will strengthen the accuracy and precision of air pollution exposure assessment, including for wildfire smoke that is becoming more prevalent. For example, one study reported on how well a model with satellite, meteorologic, and geographic/ecological inputs predicted PM_2.5_ measurements during wildfire smoke episodes at ground monitoring stations [40]. A developing opportunity for enhanced exposure assessment that can advance environmental health research is the fusion of data from low-cost sensors, reference monitors, and modeling techniques for the investigation of air pollution-related health effects [41]. Improved assessment could also lead to future improvements in forecasting, which could help the public avoid periods of exposure to hazardous air quality.

## 5. Conclusions

We found that measurements of PM_2.5_ concentrations from ground-level air monitoring stations in the Bay Area and smoke plume density scores calculated based on satellite imagery from the NOAA HMS characterized air pollution associated with the 2017–2018 wildfire seasons in California differently. Therefore, these two exposure assessment techniques commonly used in public health research and practice may convey different information about health risks associated with air quality conditions during wildfires. Incorporating data from both techniques and developing combination methods will better inform public health communication and research that aim to reduce exacerbation and incidence of disease.

## Figures and Tables

**Figure 1 ijerph-17-08164-f001:**
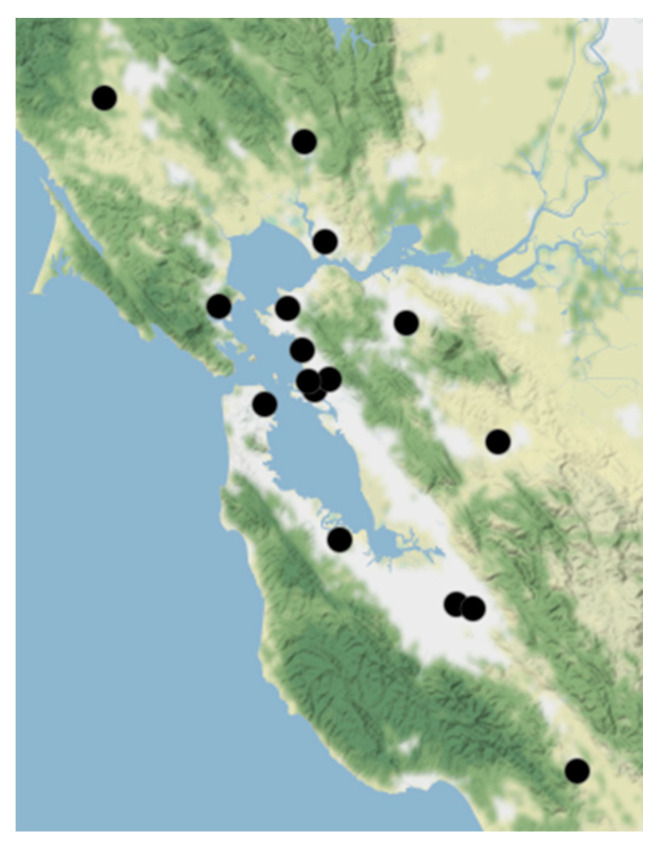
Map displaying the locations of Bay Area Air Quality Management District (BAAQMD) ground-level monitors measuring particulate matter <2.5 μm in diameter (PM_2.5_) that were included in the analysis (N = 16). (Map tiles by Stamen Design, CC BY 3.0).

**Figure 2 ijerph-17-08164-f002:**
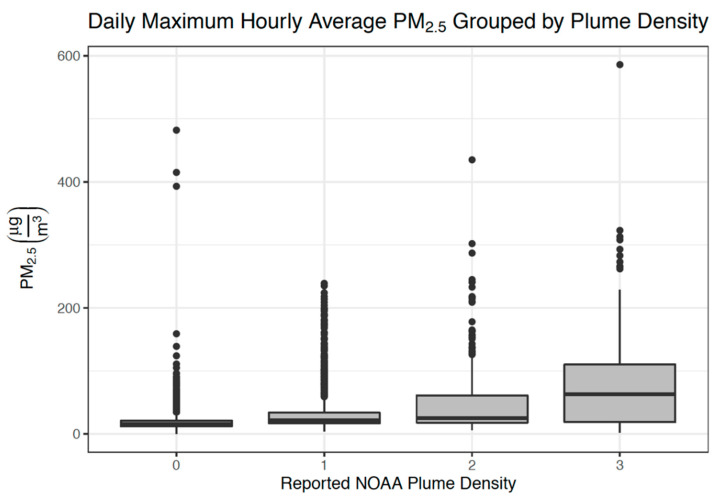
Box plot comparing ground-level PM_2.5_ concentrations and smoke plume density categories for each day in the entire 2017–2018 California wildfire seasons. The thick line within each box represents the median value, and the top and bottom of each box represent the 75th and 25th percentiles, respectively. The whiskers end at the value within the data that is no more than 1.5 times the interquartile range away from the edge of the box, and the dots beyond these whiskers are outliers.

**Figure 3 ijerph-17-08164-f003:**
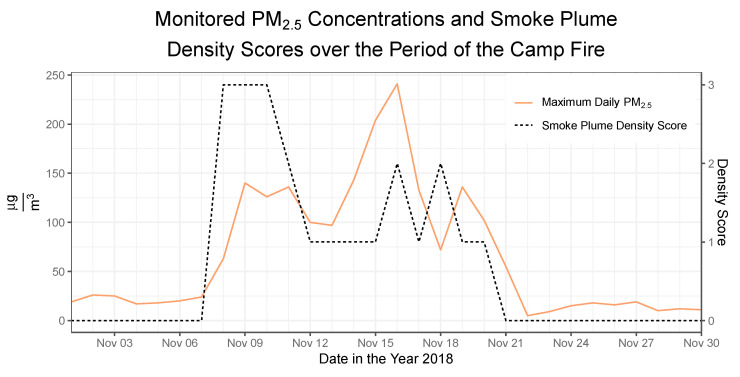
Time series plot of PM_2.5_ concentrations and smoke plume density in San Francisco during the Camp Fire.

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
