# Peer review of "Differences in the Estimation of Wildfire-Associated Air Pollution by Satellite Mapping of Smoke Plumes and Ground-Level Monitoring"

_ijerph, 2020, doi:10.3390/ijerph17218164_

Round 1
Reviewer 1 Report
The manuscript addresses an important topic and presents results from a preliminary study on the relationship of satellite smoke plume data and ground monitoring of PM2.5 for application to air quality monitoring and public health during major fire events.
The analysis is basically a case study of the San Francisco Bay area during the 2016-2017 fire seasons, in particular the more detailed analysis of the Camp Fire case study.
The paper requires major revision to address the following concerns.
The analysis should identify what conditions lead to outliers. Are particular sites prone to being outliers in Figure 2 or are all sites subject to high readings relative to smoke rating. Are there climatic conditions that can help explain the outliers – wind speed? Wind direction relative to the angle between the source of the smoke and the sensor.
The issue of PM2.5 transport is basically a 4-dimensional process, so it is understandable that the simple analysis would have some outliers. The important thing would be to identify factors that might explain the outliers.
I would suggest that the Camp Fire case study be the central focus of the paper, supplemented by the 2 year analysis. For the timeline, individual site data, or at least variability around each point on the timeline across the 16 sites is needed.
In Figure 3, the smoke rating increases first, with ~2 d lag for PM2.5. That could make sense in that the smoke plume could still be higher in the atmosphere as it first moves into the region. At the end of the 2 weeks, the smoke plume ratings decline, with a lag in PM2.5 decline. Again, the smoke moving out of the region and larger particles perhaps settled to the ground already but finer PM2.5 particles still settling to ground sensor level. Would a time series analysis perhaps help identify the time lag between the smoke plume rating which in generally occurs over a larger area and at impacted by higher altitude effects than the ground monitors.

Author Response
Thank you for the feedback. Please see the attachment for our responses.

Reviewer 2 Report
The authors have modified the manuscript based on the review comments well. However, I do not receive any response letter on which the authors should have answered my questions point by point.
Author Response
Hello,
Thank you for your comments and reviews of our manuscript. We are glad the modified manuscript addresses your previous concerns.
We apologize that you were not able to view our direct responses to your feedback. We submitted a list of reviewer comments and replies with the cover letter when submitting the updated manuscript. Here is our response to reviewer feedback that we first received:
Manuscript ID: ijerph-919465
Authors: Raj P. Fadadu, John R. Balmes, Stephanie M. Holm
We have carefully evaluated the comments provided by reviewers and have correspondingly edited our manuscript. Edits to the main text are highlighted in yellow, and below, we include the comments we received and our responses in bolded text.
All reviewers and the editor:
We received consistent feedback that the analysis we previously ran was too limited because it only included data for one ground-level monitoring station, which was located in San Francisco. Therefore, we expanded our data collection to include all the BAAQMD ground-level monitors in the San Francisco Bay Area that collected data on PM2.5 concentrations during the wildfire seasons in 2017 and 2018, which yielded a total of 16 stations, in both urban and non-urban settings (note the changes highlighted in yellow in the Methods section to reflect this). This significantly increased data for analysis and supports generalizability of results. We have modified the methods, results, and discussion sections to reflect the strengthening of our study design.
Reviewer 1:
- Figure 1 - the dark colors for plume density categories 0 and 1 make it difficult to see the exact median on the box plots
We have removed colors from the box plot, which is now Figure 2.
- Figure 2 – demonstrates the folly of trying to use an integrated measure of smoke density as a surrogate for PM2.5 concentrations. The authors suggest that this figure shows a nice correlation between the two. What they fail to mention is that the two measures differ by an order of magnitude. For example, the smoke plume density category of 3 (nominally > 22 ug/m^3) corresponds to a surface concentration of > 200 ug/m^3. Likewise, the smoke plume density category of 1 (nominally 0-10 ug/m^3) corresponds to a surface concentration of 100 ug/m^3
We have added more text to highlight the finding that air pollution assessment via ground-level monitoring and satellite imagery during the time of Camp Fire differ significantly.
- The total number of days included in this study is unclear, in part because the wildfire season is never defined. It is also unclear whether the BAAQMD site is operated on a daily basis, a 1-day-in-3 basis, or a 1-day-in-6 basis.
The time frame for data collection has been clarified (lines 84-86), and we added the total number of days (N=366, ranging from 346-366 due to some days where measurements were not available for select monitors). We state that we gathered data on daily 24-hour average and daily maximum hourly average PM2.5 concentrations from the BAAQMD sites (line 90).
Reviewer 2:
- Line 49-52, the sentence of “Many epidemiological studies have been … and healthcare utilization” has grammatical error.
We have edited the sentence to clarify the grammar.
- Line 76-80, the ground-level PM2.5 monitoring site from a Bay Area Air Quality Management District needs to be shown in the map.
We have added a figure that visualizes the locations of all 16 BAAQMD monitors on a map of the San Francisco Bay Area (Figure 1).
- Line 76-80 and Line 177-178, the authors just use one ground-level monitoring station in one city to conduct the data analysis, which is terribly unrepresentative. Consequently, the conclusion is rough and hard to believe.
We have expanded the analysis to include more BAAQMD monitors throughout the San Francisco Bay Area to be more representative (see note above).
Reviewer 3:
- I did not find information about the US EPA combining the data from satellite and ground based observations for health advisory, nor for giving health advisory based on satellite data. Could you provide more information about this?
We have edited parts of the paper discussing the EPA’s AirNow website to clarify that the AirNow website aims to provide information about fire and air quality advisories to the U.S. public (lines 74-76). In addition, it now includes both satellite and ground-based data.
- Your conclusion of course is not wrong (the discussion provides a good expose of why there are the discrepancies), but perhaps not complete as it does not provide any clues on how the information discrepancy could be handled (other than "proceed with caution"). Your conclusion as is is undermining the confidence the public needs to have in information from the authorities, as it in my opinion is not complete (no way forward).
Please consider to argue for development of combination methods (data assimilation/fusion or other methods. Some information could be found e.g. in the European efforts - CAMS, FAIRMODE, or literature on assessment of atmospheric particles. Also, the European air quality index provides some ideas how such conflicts as you demonstrate can be managed.
In conclusion, while not wrong, the message of the paper seems misleading. Your paper indicates a conflict in data which is not a right message - there are perfectly good reasons (which you list in your discussion) for the differences in the data, as each type of data provides slightly different information. I certainly agree that the current practice of exposure assessment should be improved, and you offer a practical example of this need.
We have added information at the end of the discussion that expands and clarifies the conclusions of our study (lines 221-234). We use our findings to support the development of combination methods that assimilate/fuse data for air pollution exposure assessment.
Reviewer 3 Report
I am happy with how you solved the comments; in particular I think that including 16 stations makes the results more illustrative, and I thank you for modifying the message which I feel now is appropriate.
Author Response
Thank you for your insightful comments and reviews of our manuscript. They helped us strengthen our methodology and clarify our message.
Round 2
Reviewer 1 Report
The authors have responded to the questions and comments on their manuscript and addressed them to improve the manuscript. Now this important and timely and policy-relevant manuscript will be a good contributution to the literature.
Minor point of clarification: On lines 126-128, where the targets are given for the scores 0, 1, 2, and 3, it is a bit confusing as there are only 3 categories, inplying that 1 ranges from 0-10. Isn't there a small range of values that would be counted as 0 and 1 start at some small number above 0? Could use the scoring 0= 0 to x; 1= x to 10 ; 2=....; 3=...)
Author Response
Thank you again for your feedback and review of the manuscript.
A smoke plume density score of 1 refers to PM2.5 concentrations ranging from 0 to 10 μg/m3 that can be attributable to a wildfire, and a score of 0 refers to no wildfire smoke contributing to air pollution, according to the NOAA Hazard Mapping System for Fire and Smoke Product. We have clarified this in the text of the manuscript in the Methods section (lines 104-106), when the categories are first described, and Results section (line 128).
This manuscript is a resubmission of an earlier submission. The following is a list of the peer review reports and author responses from that submission.
Round 1
Reviewer 1 Report
This manuscript performs a statistical comparison of the NOAA Hazard Mapping System Fire and Smoke product with a single, ground-based PM2.5 sensor located in San Francisco over a single fire season. Although the manuscript is well-organized and clearly presented, the ground-based site chosen for the intercomparison significantly limits the applicability and relevance of the conclusions.
The location of a ground-based PM2.5 sensor will have a huge impact on correlations with a vertically-integrated product (such as the HMS Fire and Smoke categories). Sites with the greatest likelihood of correlation will have PM2.5 concentrations dominated by smoke. This will occur at sites near active fires and at sites with significant vertical mixing. The extent of vertical mixing is controlled by the stability of the atmosphere and the amount of turbulence within the boundary layer. Sites with meteorological conditions which limit vertical mixing and sites with confounding PM sources will significantly degrade intercomparisons between the surface-based sensor and a vertically-integrated measurement. The San Francisco site suffers from both of these problems. First, San Francisco is commonly within the marine boundary layer (MBL). The MBL is typically topped by a strong temperature inversion which significantly inhibits vertical mixing. If the inversion is strong enough the MBL can be completely decoupled from the free atmosphere above. In addition, the PM2.5 sensor is located in an urban setting which is generally upwind of most of the California fires. These two factors make it likely that the PM2.5 measurements will almost always be dominated by sources other than smoke. Thus, the choice of the ground-based sensor site dictated the poor comparison with the HMS Fire and Smoke product. A more robust intercomparison would expand the scope of the study to include other urban and non-urban sites.
The manuscript makes the motivational assertion that satellite-based smoke plume data are sometimes used as a proxy for directly measured PM levels. In my experience, public health officials rely almost exclusively on in-situ, ground-based observations to sound an alert or perhaps regional chemical transport models to forecast air quality conditions on subsequent days. Proxies such as the HMS Fire and Smoke produce are more commonly used in data sparse regions without in-situ monitors. These regions are typically rural areas where the population is small. Thus, I’m not quite sure what the actual motivation for this study would be.
Minor comments/suggestions
Figure 1 - the dark colors for plume density categories 0 and 1 make it difficult to see the exact median on the box plots
Figure 2 – demonstrates the folly of trying to use an integrated measure of smoke density as a surrogate for PM2.5 concentrations. The authors suggest that this figure shows a nice correlation between the two. What they fail to mention is that the two measures differ by an order of magnitude. For example, the smoke plume density category of 3 (nominally > 22 ug/m^3) corresponds to a surface concentration of > 200 ug/m^3. Likewise, the smoke plume density category of 1 (nominally 0-10 ug/m^3) corresponds to a surface concentration of 100 ug/m^3
The total number of days included in this study is unclear, in part because the wildfire season is never defined. It is also unclear whether the BAAQMD site is operated on a daily basis, a 1-day-in-3 basis, or a 1-day-in-6 basis.
Reviewer 2 Report
Comments on the manuscript:
The manuscript entitled “Differences in the Estimation of Wildfire-Associated Air Pollution by Satellite Mapping of Smoke Plumes and Ground-Level Monitoring” compared the exposure assessment disparities between wildfire-associated air pollution and ground-level PM2.5 concentrations. This is an interesting and important work since wildfires around the word (especially in the United States, Brazil and Australia) have posed serious threats to human health and ecological environment. However, the research of the paper presented here is too simple. I suggest some additional data and analyses need to be brought in before publication:
- Line 49-52, the sentence of “Many epidemiological studies have been … and healthcare utilization” has grammatical error.
- Line 76-80, the ground-level PM2.5 monitoring site from a Bay Area Air Quality Management District needs to be shown in the map.
- Line 76-80 and Line 177-178, the authors just use one ground-level monitoring station in one city to conduct the data analysis, which is terribly unrepresentative. Consequently, the conclusion is rough and hard to believe.
- I hope that this paper is a spatial and temporal results based on a large scale.
Reviewer 3 Report
The paper addresses an issue arising when data from different sources (seemingly) lead to different messages or conclusions. Particulate matter is a difficult component to characterize in the atmosphere, and comparison of ground based measurements with satellite imagery is subject to much research.
The paper also illustrates the common difficulty in exposure assessment in how to address a combination of data from different sources that provide different elements of relevant information.
The comparison provided in the paper should lead to the conclusion that better methodologies should be developed to combine the ground-based and satellite-based PM information. Today, a number of techniques is available within air quality modelling using data fusion and data assimilation techniques to harvest information from each type of data and combine it so that maximum information is retained in the resulting product.
Questions to the authors:
1. I did not find information about the US EPA combining the data from satellite and ground based observations for health advisory, nor for giving health advisory based on satellite data. Could you provide more information about this?
2. Your conclusion of course is not wrong (the discussion provides a good expose of why there are the discrepancies), but perhaps not complete as it does not provide any clues on how the information discrepancy could be handled (other than "proceed with caution"). Your conclusion as is is undermining the confidence the public needs to have in information from the authorities, as it in my opinion is not complete (no way forward).
Please consider to argue for development of combination methods (data assimilation/fusion or other methods.Some information could be found e.g. in the European efforts - CAMS, FAIRMODE, or literature on assessment of atmospheric particles. Also, the European air quality index provides some ideas how such conflicts as you demonstrate can be managed.
In conclusion, while not wrong, the message of the paper seems misleading. Your paper indicates a conflict in data which is not a right message - there are perfectly good reasons (which you list in your discussion) for the differences in the data, as each type of data provides slightly different information. I certainly agree that the current practice of exposure assessment should be improved, and you offer a practical example of this need.
Please kindly review the paper so that the message is appropriate in light of the current views on assessment of particulate matter using data from different sources.